# PEACE: Perception and Expectations toward Artificial Intelligence in Capsule Endoscopy

**DOI:** 10.3390/jcm10235708

**Published:** 2021-12-06

**Authors:** Romain Leenhardt, Ignacio Fernandez-Urien Sainz, Emanuele Rondonotti, Ervin Toth, Cedric Van de Bruaene, Peter Baltes, Bruno Joel Rosa, Konstantinos Triantafyllou, Aymeric Histace, Anastasios Koulaouzidis, Xavier Dray

**Affiliations:** 1Endoscopy Unit, Saint Antoine Hospital, Sorbonne University, APHP, 75012 Paris, France; romain.leenhardt@aphp.fr; 2ETIS UMR 8051, CY Paris Cergy University, ENSEA, CNRS, 95000 Cergy-Pontoise, France; aymeric.histace@ensea.fr; 3Gastroenterology, Hospital de Navarra, 31008 Pamplona, Spain; ifurien@yahoo.es; 4Gastroenterology Unit, Valduce Hospital, 22100 Como, Italy; ema.rondo@gmail.com; 5Department of Gastroenterology, Skane University Hospital, Lund University, 214 28 Malmo, Sweden; ervin.toth@med.lu.se; 6Department of Gastroenterology, Ghent University Hospital, 9000 Ghent, Belgium; Cedric.vandebruaene@ugent.be; 7Klinik für Innere Medizin, Agaplesion Bethesda Krankenhaus Bergedorf, 21029 Hamburg, Germany; baltes@bkb.info; 8Department of Gastroenterology, Hospital da Senhora da Oliveira, 4835-044 Guimarães, Portugal; bruno.joel.rosa@gmail.com; 9Life and Health Sciences Research Institute, School of Medicine, University of Minho, 4704-553 Braga, Portugal; 10Hepatogastroenterology Unit, Second Department of Internal Propaedeutic Medicine, Attikon University General Hospital, Medical School, National and Kapodistrian University of Athens, 10679 Athens, Greece; ktriant@med.uoa.gr; 11Department of Social Medicine & Public Health, Faculty of Health Sciences, Pomeranian Medical University, 70-204 Szczecin, Poland; akoulaouzidis@hotmail.com

**Keywords:** small bowel capsule endoscopy, artificial intelligence, perceptions and sentiments, machine learning

## Abstract

Artificial intelligence (AI) has shown promising results in digestive endoscopy, especially in capsule endoscopy (CE). However, some physicians still have some difficulties and fear the advent of this technology. We aimed to evaluate the perceptions and current sentiments toward the use of AI in CE. An online survey questionnaire was sent to an audience of gastroenterologists. In addition, several European national leaders of the International CApsule endoscopy REsearch (I CARE) Group were asked to disseminate an online survey among their national communities of CE readers (CER). The survey included 32 questions regarding general information, perceptions of AI, and its use in daily life, medicine, endoscopy, and CE. Among 380 European gastroenterologists who answered this survey, 333 (88%) were CERs. The mean average time length of experience in CE reading was 9.9 years (0.5–22). A majority of CERs agreed that AI would positively impact CE, shorten CE reading time, and help standardize reporting in CE and characterize lesions seen in CE. Nevertheless, in the foreseeable future, a majority of CERs disagreed with the complete replacement all CE reading by AI. Most CERs believed in the high potential of AI for becoming a valuable tool for automated diagnosis and for shortening the reading time. Currently, the perception is that AI will not replace CE reading.

## 1. Introduction

Artificial intelligence (AI) is a disruptive technology, especially in medical image analysis [1,2]. AI is already assisting physicians in various clinical tasks and has shown promising results especially in digestive endoscopy [3,4,5]. To support and promote this technological innovation, the European Society of Gastrointestinal Endoscopy (ESGE) “suggests the possible incorporation of computer-aided diagnosis (CADx) to colonoscopy” [6]. Capsule endoscopy (CE) generates a very high amount of data involving gastroenterologists in a tedious task of reviewing full-length small-bowel (SB) CE videos. The use of AI in CE has already been studied in the relevant medical literature [4,7,8,9], and most authors seem optimistic and/or even ready to accept a tremendous transition. However, although AI in CE is already a commercial solution, some physicians still have difficulties accepting or some concerns related to this technology. Therefore, we aimed to evaluate the perceptions and current sentiment toward AI in CE.

## 2. Materials and Methods

An online and anonymized English survey questionnaire was designed by three gastroenterologists (AK, RL and XD). This questionnaire was distributed through GoogleForm^®^ (Mountain View, CA, USA) to European national leaders of the International CApsule endoscopy REsearch (I-CARE) group. In addition, they were asked to disseminate this online survey among their national CERs communities. Thus, participants tended to represent a diversity of training backgrounds. The survey contained 32 questions (Appendix A) divided into 6 rubrics regarding general information (physicians’ level of training and experience, practice characteristics), interest in new technologies in general, but also in endoscopy and CE, perceptions regarding the benefits of AI in endoscopy and CE, and perceptions regarding the barriers to implementing AI in CE. The term AI was defined as its general sense of “smart machines capable of performing tasks that typically require human intelligence”. Inclusion criteria for valid questionnaire returns was being a physician CER.

### 2.1. Agreement to Statements

For most questions, when addressing semi-quantitative questions, respondents were asked to rate their position on a numerical Likert response scale from 1 (strongly disagree) to 6 (strongly agree).

### 2.2. Statistical Analysis

Statistical analysis and graphical plotting were performed using Rstudio software (Version 1.2.5033, Boston, MA, USA). Qualitative variables were expressed as proportions, percentages, and raw numbers. Statistical comparisons between subgroups of responders (fellows vs. accredited gastroenterologists/below or above 40 years old, and below or above 50 years old) were performed using Chi-2 tests. Statistical significance was set at *p* < 0.05.

## 3. Results

### 3.1. Physicians’ Characteristics

A total of 380 gastroenterologists answered the questionnaire from December 2020 to March 2021 after three reminders. Most of the respondents were accredited gastroenterologist (*n* = 302; 80%), GI residents/fellows (*n* = 70; 18%), and non-medical jobs (nurse, technicians, or other healthcare providers) (*n* = 8; 2%). Overall, 20 different countries were represented. The three most active countries in terms of responders were France (*n* = 62; 16%), Spain (*n* = 56; 15%), and Italy (*n* = 45; 12%). We decided to exclude non-physician capsule readers responders (*n* = 7) from the final analysis to avoid response bias. Among responders, 333 (88%) were identified as physician CERs including 85% of accredited gastroenterologists and 15% of GI resident/fellow (Figure 1). Most CERs were between 40 and 50 years of age (*n* = 109; 33%), respectively followed by the 50–60 age group (*n* = 96; 29%), the 30–40 age group (*n* = 80; 24%), the 60–70 age group (*n* = 34; 10%), and the 20–30 age group (*n* = 14; 4%). The main practice setting was university/teaching hospital (*n* = 214; 64%), non-university hospital/district hospital (*n* = 87; 26%), and private practice (*n* = 32; 10%). The mean CE reading experience was 9.9 years (range 1–22), the current average number of CE read per year was 67 (range 2–500), and the current average number of CE readings supervised per year was 38 (range 0–500) (Figure 2).

### 3.2. Level of Information on AI

Among CERs, 38% (*n* = 126) did not consider themselves as well informed about the use of new technologies, especially informatics. Regarding CER only, 45% (*n* = 151) were aware of using any AI applications such as, for example, speech recognition, spam filters, or recommendation algorithms. Forty-six percent (*n* = 152) did not already use endoscopy AI solutions in their regular practice, but 36% (*n* = 121) planned to do so. Seventy-five percent (*n* = 251) were interested or strongly interested in receiving a generic/baseline training on AI, and 68% (*n* = 226) were interested or strongly interested in AI being part of the endoscopy training. Two-thirds (66%; *n* = 221) of responders thought that the introduction of AI would not change the relationship between physicians and patients, whereas 24% (*n* = 80) believed that this relationship would become more interactive, and 10% (*n* = 30) believed that this relationship would become more impersonal (Figure 3).

### 3.3. Perceptions of AI toward Capsule Endoscopy

Eighty per cent (*n* = 268) of CERs agreed or strongly agreed that AI will positively impact CE, and 79% (*n* = 265) agreed or strongly agreed that AI would shorten the reading time in CE. Interestingly, 71% of CERs (*n* = 236) already used quick CE reading modes in their daily practice. Moreover, 74% (*n* = 246) and 68% (*n* = 226), respectively, agreed or strongly agreed that AI would help standardize reporting in CE and characterize lesions seen in CE. Nevertheless, 71% of CERs (*n* = 236) at least moderately disagreed with the idea that AI would replace them at work. However, most CERs (68%, *n* = 227) agreed or strongly agreed that AI would allow a pre-triage of CE examinations by likelihood of being normal (Figure 4). These proportions did not vary significantly, according to the professional level (fellows vs. accredited gastroenterologists) or to the age class (below or above 40 years old, and below or above 50 years old).

Regarding CE reports, 79% (*n* = 265) of CERs disagreed or strongly disagreed that AI would allow automated reporting without validation by the gastroenterologist. Regarding the time limit to get the results from an AI solution for CE reading, the majority of CERs (59%; *n* = 196) believed that a few minutes would be reasonable, whereas 29% (*n* = 96) would tolerate a few hours. Moreover, only 44% of CERs (*n* = 157) believed that AI would increase the number of requests for CE examinations. Finally, the maximum mean number of false positive frames per examination that CER would be ready to review was 222 (ranging from 50 to 1000).

### 3.4. Perceptions Regarding the Barriers for AI Implementation

Sixty percent of CR (*n* = 201) at least moderately agreed that AI implementation could lead to operator dependence. Although 51% of CER (*n* = 170) thought that patients should be aware of the use of AI in CE reading, 53% (*n* = 175) of CER believed that patients should not expressly consent to AI in CE reading. Regarding who should take the responsibility of AI-based reporting in CE, 75% of CERs (*n* = 250) believed that the responsibility remains at least with the endoscopist. In contrast, the other CERs believed that responsibility should be shared between the clinician requesting the exam, the developers of AI applications, and insurance companies. Finally, regarding the cost of AI implementation in CE, 65% (*n* = 218) of CERs agreed or strongly agreed that this concern is of significant importance (Figure 5).

## 4. Discussion

This study reports the first evaluation on perception and attitudes toward AI in CE involving a large group of international CERs. This survey revealed that a majority of CERs have confidence in the implementation of AI applications in CE. Our results indicate that most of the responders are interested in undergoing a learning program on AI. Moreover, 92% of CERs agree that AI should be part of the regular endoscopy training. These perceptions are per the survey of Wadhwa et al. [10], in which 86% of physicians reported interest in AI-assisted colonoscopy. Further illustrations, especially in radiology, revealed the same general enthusiasm regarding AI. For example, Waymel et al. [11] showed through an electronic survey sent to radiologists that 79.3% (*n* = 214) of respondents think AI will positively impact their future practice. Moreover, in the study of Pinto Dos Santos et al., 77% and 86% of radiologists and students, respectively, agreed that AI would revolutionize and improve radiology [12]. These data corroborate the important expectation of AI implementation in medicine.

One of the primary functions of AI in CE is to reduce reading times while maintaining the highest possible sensitivity for abnormalities detection. The task of reviewing full-length SBCE reading is tedious, and CERs spend on average 30 to 120 min to review and interpret complete SBCE examinations encompassing thousands of images [13]. Moreover, Beg et al. showed that reader accuracy declines after reviewing one full-length SBCE recording and that accuracy is also influenced by reading speed [14]. In 2019, a large multicenter study in China showed that an AI-based algorithm could reduce the mean CE reading time to 6 min per video while maintaining a high sensitivity for abnormalities detection [4]. Although the ESGE guidelines recommend that quick CE reading modes “may be used to scan the small bowel for diffuse lesions (…) but should not be relied on to detect an isolated lesion” [15], our results interestingly report that 71% of CER (*n* = 236) already use quick CE reading modes in their daily practice, demonstrating a genuine desire to reduce CE reviewing time. Moreover, some could hypothesize that AI implementation would lead to an increase in CE requests, but 53% of CR (*n* = 176) suggest that no significant change is to be expected.

Regarding the potential cost of AI implementation in CE, 65% (*n* = 218) of CER agreed or strongly agreed that it is a topic of major importance. Most CE manufacturers are currently working on developing AI-based solutions for CE reporting. However, the medical literature remains scarce on the potential performances of these devices, as most available studies are retrospective, using local datasets without any external validation. These drawbacks may hamper the cautious implementation of this novel technology. Thus, evaluating the cost-effectiveness of AI solutions in CE is needed to better determine the feasibility and safety of this technology in daily practice and whether it should be implemented as a first or second reader [16]. The specific question of a cost-effectiveness threshold was not addressed in this study and should be considered as a perspective for further medico-economic studies.

Our study has several strengths. First, it is the largest survey addressing gastroenterologists’ perceptions of AI toward CE. Second, this survey included various modalities of medical practice (university/teaching hospital, non-university, and private practice). Third, all participants were chosen to represent a diversity of training backgrounds. Nevertheless, we must also acknowledge a few limitations. First, the very real possibility of participation bias: CERs who voluntarily answered this survey might be interested in this topic, especially academic physicians (63%) who were a priori more prone to answering. The authors hypothesize that this higher participation could probably be due to network impact and can be responsible for participation bias. However, neither age nor professional level influenced the perceptions of AI toward CE, meaning that CERs share the same general optimism regarding AI implementation in CE. Second, our online survey did not lead to a debate, and a physical meeting would likely have enriched the discussion rather than an online survey. Third, the term “AI” was given as its general sense of “smart machines capable of performing tasks that typically require human intelligence”. Then, potential benefits of AI in CE were listed (“shorten reading time”, “help characterizing”, “help standardizing”, “automated reporting”…) and given to the participants in their general senses (rather than “reading in less than 10 min” or “detecting with sensitivity over 95%”, etc.). We decided so, first because standards are not defined here yet (although some research groups are working on it), second because any precise threshold may influence results where we aimed to measure qualitative outcomes (sentiments, perceptions, expectations) in the community of CER. A review on AI in SBCE from Dray et al. [9] has listed all technological bricks under development and evaluation. Finally, the authors could not calculate a participation rate, as invitations to answer this survey were spread by e-mail throughout many of countries by corresponding capsule leaders in their community of capsule readers; this could have participated in a non-response bias.

Some initiatives have focused on research questions trying to identify priorities related to AI. There is a general will for collaborative efforts to guide better and approach the barriers of AI implementation. In their paper, Vinsard et al. [17] have proposed some fundamental principles for AI system development and clinical testing to promote quality assurance of CAD and diagnosis in colonoscopy. Another example from the study of Ahmad et al. illustrates this trend as they also proposed a systematic process through a modified Delphi methodology to identify research priorities for AI in colonoscopy [18]. All these considerations regarding AI implementation have led to a current ongoing work from our group to map out research, clinical gaps, and challenges for AI in CE with some CE international leaders.

Some ethical issues also must be discussed regarding the legal aspects of AI implementation for CE reading. Depending on how AI is used, the delegation of diagnostic tasks to machines is tricky and unanswered. AI tools have demonstrated their capacity to enhance physicians’ performance detecting colonic polyps or Barrett’s esophagus, but AI use introduces the risk of influencing physicians’ decisions. Our result illustrates that 60% of CR (*n* = 201) at least moderately agreed that AI implementation could lead to operator dependence. One thing is certain; the gastroenterologist community must contribute to the AI development for better application in clinical practice. General optimism is needed for attracting funding in AI algorithms development and for their wide adoption. Indeed, AI in CE will probably be an “adversarial” rather than “assistive” tool meaning that, as for radiologists, only selected frames will be proofread by the CER, while others will simply not be double-checked.

Moreover, human trust in AI is of tremendous importance to support its implementation. Still, AI continues to frighten a part of our society, as AI algorithms can have black boxes even to their creators. AI algorithms make decisions and predictions as humans do without being able to describe their way of operating. More transparency on these black boxes will widen its adoption. However, the community of CERs must be prepared, as AI use in CE will automatically lead to inevitable mistakes. In the new era of AI, these risks could be compared to drug accidents, and it should not slow down AI development and obscure all the potential of this new technology. Indeed, some authors already promote creating a reliable system of AI techno vigilance to solve all these issues [19] and maybe define strict liabilities for harm inflicted by AI.

## 5. Conclusions

Capsule readers showed general enthusiasm toward AI in CE. The majority of CR believe in the high potential of AI for shortening reading times and for the semi-automated detection of abnormalities. Although CR cannot yet rely entirely on such solutions, before high-quality trials demonstrate that this approach is highly sensitive and cost-effective, the community seems eager to welcome the technology.

## Figures and Tables

**Figure 1 jcm-10-05708-f001:**
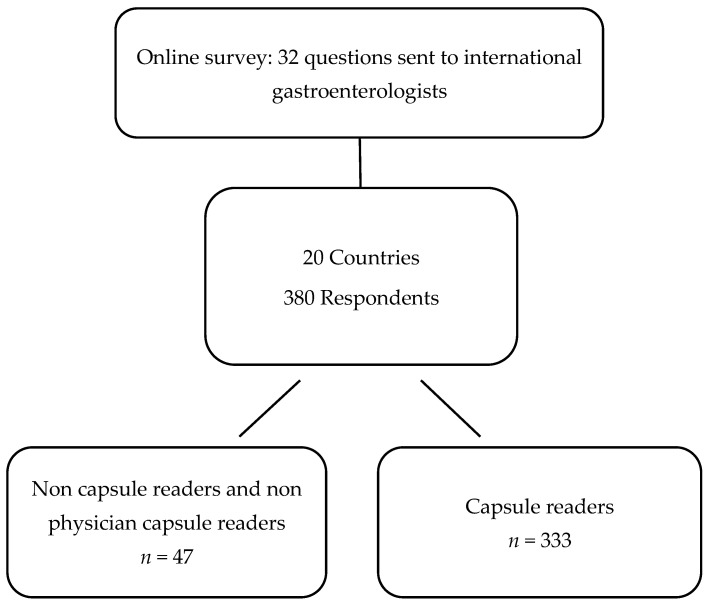
Flowchart of the study.

**Figure 2 jcm-10-05708-f002:**
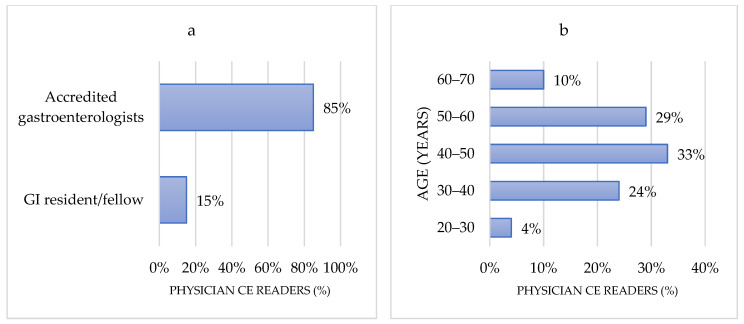
Capsule readers characteristics: (**a**) Current position of physician CE readers; (**b**) Age distribution of capsule readers; (**c**) Survey responders by country (Physician capsule readers only); (**d**) Main practice setting. CE: Capsule endoscopy; GI: Gastrointestinal.

**Figure 3 jcm-10-05708-f003:**
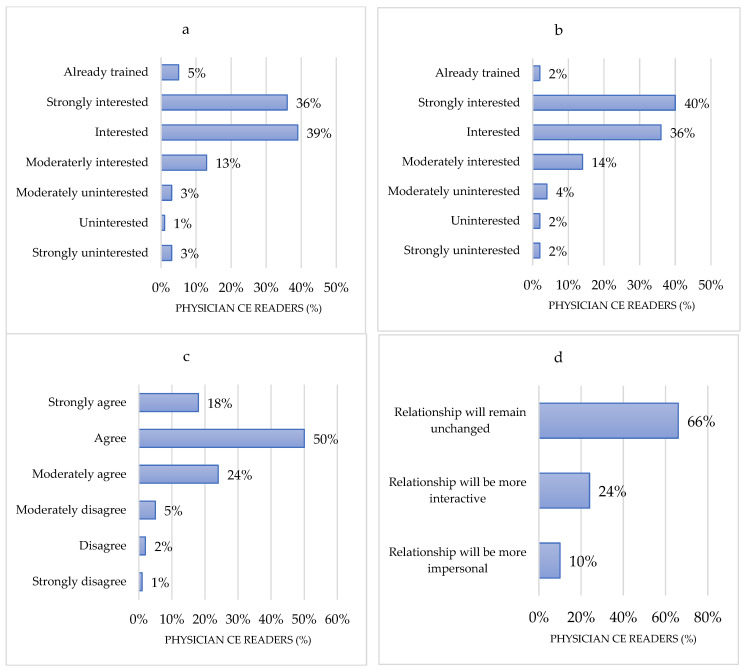
Level of information of capsule readers on artificial intelligence in general and in medicine. (**a**) How interested are you in receiving a generic/baseline training on AI?; (**b**) How interested are you in receiving a technically advanced training on AI?; (**c**) AI should be part of the endoscopy training; (**d**) How will the relationship between the endoscopist and the patient change with the introduction of AI?

**Figure 4 jcm-10-05708-f004:**
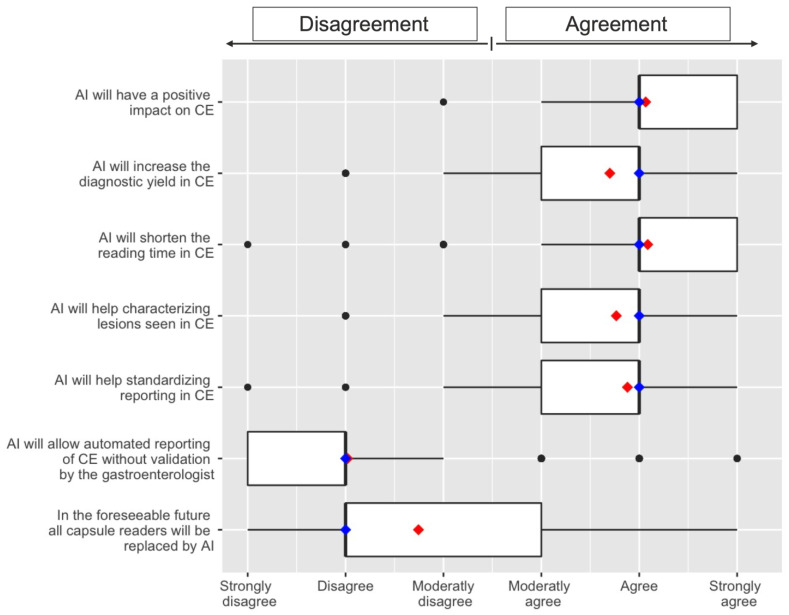
Perceptions of AI toward capsule endoscopy. Box plots show the distribution of responses using a Likert scale ranging from 1 to 6 (strongly disagree to strongly agree). The mean value is plotted with a red dot. The vertical blue bar represents the median value; 25th and 75th percentiles are set as box limits.

**Figure 5 jcm-10-05708-f005:**
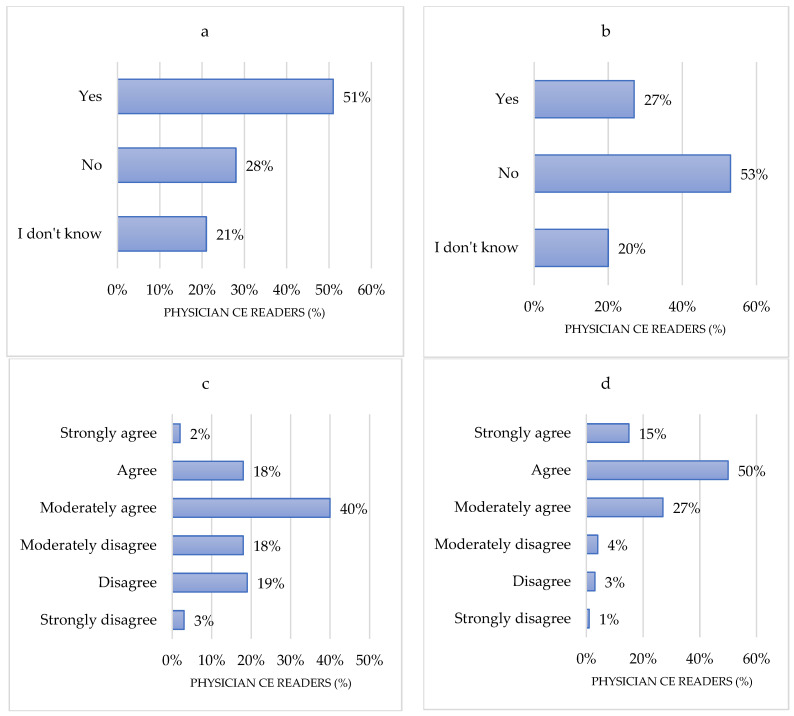
Perceptions regarding the barriers for AI implementation. (**a**) Patients should be aware about the use of AI in CE reading; (**b**) Patients should specifically consent to the use of AI in CE reading; (**c**) AI in CE will lead to operator dependence on the technology; (**d**) Cost is important in implementing AI in CE reading. CE: Capsule endoscopy.

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
