# Peer review of "PEACE: Perception and Expectations toward Artificial Intelligence in Capsule Endoscopy"

_jcm, 2021, doi:10.3390/jcm10235708_

Round 1

Reviewer 1 Report

This is the report of the results of an online and anonymized English survey questionnaire on perception and expectations towards Artificial Intelligence (AI) in Capsule Endoscopy (CE) amongst European gastroenterologists. The subject of the study is of high interest as acceptance of new technology is a major part of its successful application.

The survey distribution method seems adequate; the manuscript is well written and comprehensive. There are some remarks and questions remain open:

Major points

  • ‘AI‘ seems as big a technology to bring major transformation into the field of CE as the authors indicate. However, a major drawback of the manuscript in the current form is the lack of any precise definition of AI in CE; what do
    • The authors correctly mention that ‘AI’ is integral part of some commercially available software tools already today; however, they seem not to differentiate these from future tools that are currently under scientific evaluation. It seems essential to detail the different kinds of AI-tools in CE within this report (e.g., structured analysis support, imaging analysis to detect pathological findings, CAD/CADe/CADx, lesion recognition vs. characterization). It would help the understanding to include this information in the background/introduction and methods and to discuss different AI applications with possible impact on the understanding of the interviewee and the survey’s results.
    • What exactly were the interviewee asked with respect to the definition of ‘AI in CE’?
    • Material and Methods: To analyze the potential “benefits of AI in endoscopy and CE” those have to be defined and assessed with qualitative and/or quantitative analysis

Minor issues

  • There seems to be an imbalance towards doctors from University/teaching hospitals. Possible influence on the survey results should be discussed in detail.
  • It remains unclear how many gastroenterologists were reached in total, i.e. the survey response rate is not reported, however should be mentioned to account for non-response bias.
  • The authors mention ‘a physical meeting’; has there been a planned meeting for discussing the survey? Please explain.
  • Line 67: “Inclusion criteria for valid questionnaire returns were being a CER and a physician.” Does this mean, that both criteria had to be fulfilled or any of both?

Author Response

  • AI‘ seems as big a technology to bring major transformation into the field of CE as the authors indicate. However, a major drawback of the manuscript in the current form is the lack of any precise definition of AI in CE;

The term ”AI” was given as its general sense of “smart machines capable of performing tasks that typically require human intelligence” for the first questions. This general definition is now given in the “Material and Methods section” of our paper (page 3). Then, further specific questions, focused on various tasks (detecting abnormal frames, detecting regions of interest within abnormal frames, characterizing, placing landmarks, rating cleanliness, reporting). These details are now added in the “Material and Methods” section of the paper (page 3), and a supplementary document with a list of questions is provided (see answer to query 3)

  • The authors correctly mention that ‘AI’ is an integral part of some commercially available software tools already today; however, they seem not to differentiate these from future tools that are currently under scientific evaluation. It seems essential to detail the different kinds of AI tools in CE within this report (e.g., structured analysis support, imaging analysis to detect pathological findings, CAD/CADe/CADx, lesion recognition vs. characterization). It would help the understanding to include this information in the background/introduction and methods and to discuss different AI applications with possible impact on the understanding of the interviewee and the survey’s results. 

Please see answer to query 1 and supplementary material. We have also add a reference to a most recent review of the literature by our group, addressing those various tasks and perspectives (Dray et al. JGH).

  • What exactly were the interviewee asked with respect to the definition of ‘AI in CE’? .

Regarding the reviewers comments, the authors only decided to mention the 32 questions for which answers were described in the manuscript. Thus, minor corrections were done for the Flowchart and an appendix with all 32 questions has been proposed for the manuscript revision. The authors will let to the editor the possibility to add it or not to the final manuscript.

  • Material and Methods: To analyze the potential “benefits of AI in endoscopy and CE” those have to be defined and assessed with qualitative and/or quantitative analysis.

Thank you for this comment. As explained in answers to queries 1 to 3, the potential benefits of AI in CE were listed (“shorten reading time”, “help characterizing”, “help standardizing”, “automated reporting”…) and given to the participants in their general senses (rather than “reading in less than 10 minutes” or “detecting with sensitivity over 95%”, etc). We decided so, first because standards are not defined here yet (although our group and others are working on it), second because any precise threshold may influence results where we aimed to measure qualitative outcomes (sentiments, perceptions, expectations) in the community of CE readers. Regarding the study’s methodology, a semi-quantitative assessment was performed using a numerical Likert scale varying from 1 to 6 for most answers. This scale proposed an even (6) number of possible answers, thus reducing the option for responder to give “mean” answers.

Minor issues

  • There seems to be an imbalance towards doctors from University/teaching hospitals. Possible influence on the survey results should be discussed in detail. 

We agree with the reviewer’s comment. Please find the following changes on page 11. « CERs who voluntarily answered this survey might be interested in this topic, especially academic physicians (63%) who were a priori more prone to answering. Authors hypothesize that this higher participation could probably be due to network impact and can be responsible for participation bias. »

  • It remains unclear how many gastroenterologists were reached in total, i.e. the survey response rate is not reported, however should be mentioned to account for non-response bias. 

The authors thank the reviewer for this comment. Unfortunately, the authors do not have this information as invitations to answer this survey were spread by e-mail throughout many of countries by corresponding capsule leaders in their community of capsule readers. Thus, the authors were not able to calculate a survey response rate. We acknowledge in the discussion (page 11) that it is a potential bias.

  • The authors mention ‘a physical meeting’; has there been a planned meeting for discussing the survey? Please explain. 

The authors apologize for this misunderstanding. No physical meeting was planned. Instead, the authors discussed the potential bias of such a study’s methodology based on internet questionnaires. In the new version of the manuscript (page 11) the authors wrote that « a physical meeting would likely have enriched the discussion rather than an online survey.»

  • Line 67: “Inclusion criteria for valid questionnaire returns were being a CER and a physician.” Does this mean, that both criteria had to be fulfilled or any of both? 

Inclusion criteria included being both a CER and a physician. Only a few non physicians capsule readers answered the questionnaire (n=7) and authors decided to remove them from the final analysis in order to avoid bias and homogenize the results. Corrections have been made accordingly in the manuscript (page 4, line 126): ”Inclusion criteria for valid questionnaire returns was being a physician CER.”

Reviewer 2 Report

This study assessed about the current perception and expectations of experts toward AI in capsule endoscopy. The topic of this study was very interesting and the result was informative. I have some questions.

  1. Authors indicated that there is no difference in the results of the questionnaire between capsule readers in university / teaching hospital and those in non university hospital/ district hospital, but it would be nice if authors could explain what the results were in terms of training or relationship with the patient, and if possible, show them in a figure.
  2. The cost-effectiveness is also likely to be important factor in applying AI in capsule endoscopy. If possible, it would be helpful to explain the cost at which AI can be actively applied to capsule endoscopy.
  3. The time saving is also very important in the interpretation of capsule endoscopy. It would be good to suggest how much time reduction would be acceptable in the application of AI in capsule endoscopy.
  4. Who will be held responsible if there is a legal problem in the diagnostic process when applying AI in capsule endoscopy? Is it a company or a doctor? There is a need to discuss the limits of this legal liability.

Author Response

This study assessed the current perception and expectations of experts toward AI in capsule endoscopy.

The topic of this study was very interesting and the result was informative. I have some questions.

  • Authors indicated that there is no difference in the results of the questionnaire between capsule readers in university / teaching hospital and those in non university hospital/ district hospital, but it would be nice if authors could explain what the results were in terms of training or relationship with the patient, and if possible, show them in a figure. 

We can only describe this comparison between university vs non-university settings, as the background of physicians and the referral of patients may be different. We do not have any data on the training and clinical involvement of CER physicians related to those settings. We would rather not mention further interpretation that would therefore be speculative.

  • The cost-effectiveness is also likely to be important factor in applying AI in capsule endoscopy. If possible, it would be helpful to explain the cost at which AI can be actively applied to capsule endoscopy. 

The authors thank the reviewer for this relevant comment, as one of the main goals of AI in CE is to reduce the reading time (and therefore cost). We have addressed the question of reducing reading time. Sixty-five percents of participants agreed or strongly agreed that the cost issue is important (figure 5). The specific question of a cost-effectiveness threshold was not addressed. It is now mentioned in the discussion as a perspective for further medico-economic studies (page 10). It is also mentioned as “a complex issue as AI would decrease direct cost (reading and interpretation), decrease some indirect costs (length of hospital stay for example), but may increase other indirect costs (due to possibly more CE to be prescribed, for example).”

  • The time saving is also very important in the interpretation of capsule endoscopy. It would be good to suggest how much time reduction would be acceptable in the application of AI in capsule endoscopy. 

The authors thank the reviewer for this comment. The study questionnaire brings exciting information regarding the time limit to get an AI solution for CE reading. Most CERs (59%; n=196) believe that a few minutes would be reasonable, whereas 29% (n=96) would tolerate a few hours. Moreover, this study adds additional interesting data regarding the maximum mean number of false-positive frames per examination that CER would be ready to review (n=222). We decided to leave the questions of thresholds (precise performance criteria) opened, when collecting semi-quantitative data on perceptions and expectations.

  • Who will be held responsible if there is a legal problem in the diagnostic process when applying AI in capsule endoscopy? Is it a company or a doctor? There is a need to discuss the limits of this legal liability. 

The authors thank the reviewer for these very challenging, but unsolved, problems. The legal aspect is a hot topic with the arrival of AI tools (not only in medicine, but for autonomous vehicles as well for example). AI is a new technology that is not totally implemented in our daily practice, so there is no available nor generic answer to these questions.

This survey did not intend to answer these questions, but to measure how end-users (CER) foresee their own, future, responsibility in this setting. Responders were asked who should take the responsibility of AI-based reporting in CE. Seventy-five per cent of them (n=250) believed that the responsibility remains with the endoscopist. In contrast, the other CERs believe that responsibility should be shared between the clinician requesting the exam, the developers of AI applications and insurance companies.

Overall, we have mentioned at the end of the discussion (page 12), that these questions are crucial for medical imaging (of all kind). Indeed, AI in CE will probably be an “adversarial” rather than “assistive” tool (meaning that, as for radiologists, only selected frames will be proofread by the CER while other will be simply not be double-checked). Plus, we comment (page 12) that some authors already promote creating a reliable system of AI techno-vigilance to solve all these legal issues and maybe define strict liabilities for potential harm inflicted by missed AI-diagnosis.